# Comparing the Efficacy of Two Generations of EGFR-TKIs: An Integrated Drug–Disease Mechanistic Model Approach in EGFR-Mutated Lung Adenocarcinoma

**DOI:** 10.3390/biomedicines12030704

**Published:** 2024-03-21

**Authors:** Hippolyte Darré, Perrine Masson, Arnaud Nativel, Laura Villain, Diane Lefaudeux, Claire Couty, Bastien Martin, Evgueni Jacob, Michaël Duruisseaux, Jean-Louis Palgen, Claudio Monteiro, Adèle L’Hostis

**Affiliations:** 1Novadiscovery SA, Pl. Giovanni da Verrazzano, 69009 Lyon, France; hippolyte.darre@novadiscovery.com (H.D.); claudio.monteiro@novadiscovery.com (C.M.); 2Respiratory Department and Early Phase, Louis Pradel Hospital, Hospices Civils de Lyon Cancer Institute, 69100 Lyon, France; michael.duruisseaux@chu-lyon.fr; 3Cancer Research Center of Lyon, UMR INSERM 1052 CNRS 5286, 69008 Lyon, France; 4Université de Lyon, Université Claude Bernard Lyon 1, 69100 Lyon, France

**Keywords:** lung adenocarcinoma, EGFR-TKI, precision oncology, in silico, computational oncology

## Abstract

Mutationsin epidermal growth factor receptor (EGFR) are found in approximately 48% of Asian and 19% of Western patients with lung adenocarcinoma (LUAD), leading to aggressive tumor growth. While tyrosine kinase inhibitors (TKIs) like gefitinib and osimertinib target this mutation, treatments often face challenges such as metastasis and resistance. To address this, we developed physiologically based pharmacokinetic (PBPK) models for both drugs, simulating their distribution within the primary tumor and metastases following oral administration. These models, combined with a mechanistic knowledge-based disease model of *EGFR*-mutated LUAD, allow us to predict the tumor’s behavior under treatment considering the diversity within the tumor cells due to different mutations. The combined model reproduces the drugs’ distribution within the body, as well as the effects of both gefitinib and osimertinib on EGFR-activation-induced signaling pathways. In addition, the disease model encapsulates the heterogeneity within the tumor through the representation of various subclones. Each subclone is characterized by unique mutation profiles, allowing the model to accurately reproduce clinical outcomes, including patients’ progression, aligning with RECIST criteria guidelines (version 1.1). Datasets used for calibration came from NEJ002 and FLAURA clinical trials. The quality of the fit was ensured with rigorous visual predictive checks and statistical tests (comparison metrics computed from bootstrapped, weighted log-rank tests: 98.4% (NEJ002) and 99.9% (FLAURA) similarity). In addition, the model was able to predict outcomes from an independent retrospective study comparing gefitinib and osimertinib which had not been used within the model development phase. This output validation underscores mechanistic models’ potential in guiding future clinical trials by comparing treatment efficacies and identifying patients who would benefit most from specific TKIs. Our work is a step towards the design of a powerful tool enhancing personalized treatment in LUAD. It could support treatment strategy evaluations and potentially reduce trial sizes, promising more efficient and targeted therapeutic approaches. Following its consecutive prospective validations with the FLAURA2 and MARIPOSA trials (validation metrics computed from bootstrapped, weighted log-rank tests: 94.0% and 98.1%, respectively), the model could be used to generate a synthetic control arm.

## 1. Introduction

Lung cancer remains the leading cause of cancer-related deaths globally, with over 1.8 million deaths and 2.2 million new cases yearly [1]. The high mortality rate is largely attributed to late-stage diagnosis despite advances in screening [1]. Non-small-cell lung cancer (NSCLC) constitutes 80% of cases, with about 40% being lung adenocarcinoma (LUAD) [2]. Driver mutations, like the mutation of the epidermal growth factor receptor (*EGFR*) gene, prevalent in 48% of Asian and 19% of Western LUAD patients, play a crucial role in tumor growth and invasiveness [3]. *EGFR* mutations, commonly, either exon 21 L858R insertion or deletion on exon 19, trigger uncontrolled activation of the EGFR receptor, promoting cancer cell growth and survival [4]. This understanding has led to targeted therapies focusing on these driver mutations, offering an alternative to traditional chemotherapy. These therapies specifically target driver mutations and disrupt their ability to drive tumor proliferation and metastasis apparition. EGFR tyrosine kinase inhibitors (TKIs) compete with ATP at the intracellular kinase domain of EGFR, inhibiting its auto-phosphorylation and constitutive downstream pathways’ activation. Since 2015, several EGFR-TKIs have been approved by the Food and Drug Administration (FDA) to treat LUAD patients with an *EGFR* driver mutation. The first-generation TKIs, namely, erlotinib and gefitinib, are reversible inhibitors, meaning that once the drug unbinds its target, the effects of the drug disappear, contrary to second-generation TKIs afatinib and dacomitinib, which are irreversible inhibitors. Osimertinib, a third-generation, irreversible TKI, was later developed to overcome resistance to first- and second-generation EGFR-TKIs. It targets both common *EGFR* mutations and the T790M resistance mutation, an *EGFR* mutation on exon 20 that frequently appears during treatment with first- or second-generation TKIs and causes resistance to these treatments [5]. Despite these advancements in targeted therapies, the prediction of patient response to specific EGFR-TKIs and the emergence of resistance to treatment pose significant challenges [6]. This uncertainty in treatment response highlights a gap in the ability to make informed decisions regarding the design of new drugs. This article aims to shed light on a possible way to use the existing knowledge to support clinical decisions. Specifically, we explore the potential of developing clinical predictive models. These models could assist in evaluating efficacy and/or resistance profiles of existing and upcoming EGFR-TKIs. By incorporating tumor heterogeneity, mechanistic models can help identify patients most likely to benefit from specific targeted therapies. They can also improve the exploration of regimen strategies to overcome or delay the development of resistance. These models could take personalized medicine a step further by providing an additional tool for patient selection in clinical trials. They could facilitate the individual optimization of therapeutic interventions, potentially improving treatment outcomes. Therefore, they could become an invaluable tool for the planification of *EGFR*-mutated LUAD care strategies. Through the application of a knowledge-based model (KBM), this article proposes an avenue for solutions [7]. The KBM presented in this article can simulate the responses of a specific population to different TKIs targeting the same receptor, thereby enabling comparative analysis of various treatment strategies and sequences. To achieve this, we refine a KBM named ISELA-V1 that we previously developed [8], and which mechanistically represents the main biological phenomena related to EGFR-mutated LUAD disease, and combine it with physiologically based pharmacokinetics models of osimertinib and gefitinib that incorporate data on these two EGFR-TKIs, their mode of actions and their specific resistance mechanisms. Using this updated model, named ISELA-V2, we simulate responses to both EGFR-TKIs in the same population and compare the outcomes and efficacy of both treatments. This article intends to highlight the potential of mechanistic modeling for optimization of therapeutic interventions, which would be even further improved if given individual patient characteristics, paving the way for the future development of tools that could enhance decision-making in the context of introducing a new drug to market.

## 2. Materials and Methods

### 2.1. Improvement of ISELA-V1 Model

In this paper, we aimed to simulate responses to both gefitinib and osimertinib within the same population to compare their outcomes and efficacy. To achieve this, significant modifications were made to the ISELA-V1 model, originally presented by L’Hostis et al. [8], to develop a more sophisticated disease model. This refined model represents neoangiogenesis, the cell cycle and cell death, as well as metastasis evolution, to consider the impact of EGFR-TKIs on tumor growth while more precisely considering cancer hallmarks. All these additions are presented in the Appendix A. The combination of the improved disease model with gefitinib and osimertinib treatment options has been named ISELA-V2. Its outputs are notably composed of the time to progression as defined by the RECIST criteria version 1.1 [9].

### 2.2. Modeling of Osimertinib and Gefitinib

#### 2.2.1. Characteristics of the Two Epidermal Growth Factor Receptor Tyrosine Kinase Inhibitors and Modeling Strategy

Osimertinib and gefitinib are both small molecules. They penetrate inside tumor cells, bind to the intracellular domain of the EGFR and inhibit its phosphorylation by ATP. They thereby prevent the activation of downstream signaling pathways, leading to cell growth and survival [10]. However, due to their distinct physicochemical properties, the two drugs have specific characteristics that result in different clinical results for the two treatments. These differences include the distribution of the drug in the organism, the differences in affinity between the receptor and the drug, differences in the reversibility and irreversibility of bindings with the EGFR for gefitinib and osimertinib, respectively, their mechanisms of resistance and the off-target bindings responsible for adverse effects. The ISELA-V2 model notably aims to prove that, by modeling these differences and with the same disease model, it is possible to simulate the observed differences in terms of clinical results for the two treatments. The modeling of these differences is detailed in the following paragraphs, with the exception of the off-target binding and adverse effects, which were out of the scope of the study and constitute a limitation of the model.

#### 2.2.2. Physiologically Based Pharmacokinetic Model of Osimertinib and Gefitinib

To assess the concentration of the TKI present in the tumor tissues contributing to reduced growth signals, we developed two PBPK models, one for gefitinib and one for osimertinib. PBPK models are designed based on known physiology. They predict how a drug will be absorbed, distributed, metabolized and eliminated (ADME) by the body depending on its physicochemical properties. These models estimate the concentration of the drug in all the main organs of the body, thereby making them especially relevant for the modeling of multiple tumor sites. Furthermore, the model structure, although complex, is similar for two drugs with analogous physicochemical properties. For this reason, the PBPK models of gefitinib and osimertinib were built using the same equations. Nonetheless, a particularity of osimertinib is that one of its metabolites, AZ5104, also acts as an EGFR-TKI [11]. The metabolization of osimertinib to AZ5104 in the liver has therefore been modeled, as well as the equations of distribution of AZ5104 from the liver to different tissues and tumors. The structure of the PBPK models with the equations and underlying assumptions is presented in more detail in the Appendix A. The equations of the PBPK models regulate the pharmacokinetics of the drugs and grant access to the concentration of the drug in each of the main organs of the body. For the distribution of the drug in the tumors (primary tumor or metastasis), it is assumed that the concentration of drug in the tumor is directly proportional to its concentration in the organ in which the tumor is located. Quantitative public data on osimertinib and gefitinib mechanisms of action are mainly available for mice studies. Therefore, a mouse setting was included in the model, keeping all the equations but changing all the physiological parameters to correspond to a mouse. The parameters to calibrate related to the pharmacokinetics of the drugs were calibrated separately in mice and humans to reproduce the plasmatic profile of the drug after administration in each species.

#### 2.2.3. Modeling Tyrosine Kinase Inhibitor Mechanism of Action

The disease model has been developed in the context of *EGFR*-mutated LUAD, and the modeling of the growth factor signaling, including the signals arising from EGFR activation, follows the activation pathways depicted in Figure 1. Once activated, the phosphorylation of EGFR activates the MAPK and PI3K/AKT downstream pathways, eventually leading to cell proliferation and survival. Not all intermediaries have been modeled; the first modeled actors are RAS and PI3K. It is of note that several other tyrosine kinase receptors that activate the same downstream pathways as EGFR exist [12]. Among them, we chose to model MET and VEGFR receptors, as they play specific roles in the phenomena of interest in the context of the use of the model. The resistance of a tumor to EGFR-TKI treatment can be caused by an amplification of the MET receptor [6], and the VEGFR is a receptor that plays a role in the neoangiogenesis of the tumor [13]. The activation of RAS and PI3K by EGFR, VEGFR and MET is, respectively, depicted in Equations (Equation 1) and (Equation 2).
(1)dRasActivedt=RasInactive×(kaRasByMet×cMetActive=+kaRasByVegfr×VegfrActive=+kaRasByEgfr×EgfrActive×propPhosphoEgfrActive)−RasActive×kdaRas
(2)dPi3kActivedt=Pi3kInactive×(kaPi3kByMet×cMetActive=+kaPi3kByVegfr×VegfrActive=+kaPi3kByEgfr×EgfrActive×propPhosphoEgfrActive=+kaPi3kByRas×RasActive)−Pi3kActive×kdaPi3k

Table 1 presents the parameters used in Equations (Equation 1) and (Equation 2); they were parameterized based on an MAPK cascade model developed by Schoeberl et al. [14].

The model represents the different states of EGFR: inactivated, activated and phosphorylated. The parameter propPhosphoEgfrActive corresponds to the proportion of activated EGFR that is phosphorylated. It multiplies the concentration of activated EGFR to model the concentration of EGFR activating downstream pathways. It includes the mechanism of action of tyrosine kinase inhibitors that impacts the strength of the signal by competing with ATP in the intracellular domain and preventing the phosphorylation of the receptor. It is defined by: (3)$$propPhosphoEgfrActive=\frac{\left[ATP\right]}{\left[ATP\right]+atpAffinity\times \left(1+\frac{1+[TKI]}{Ki_{TKI}}\right)}$$

The ATP affinity with the EGFR intracellular domain can differ across EGFR mutation types, as well as the inhibition coefficient of the tyrosine kinase inhibitor [15,16,17]. The values of the inhibition coefficient are given in Table 2.

As expected, the *T790M* mutation that causes resistance to gefitinib treatment is linked to a high $Ki_{TKI}/atpAffinity$ ratio for gefitinib compared to other sensitive mutations. The exon 20 resistant mutation that causes resistance to both gefitinib and osimertinib also presents a high $Ki_{TKI}/atpAffinity$ ratio for gefitinib and osimertinib. The disease model, for which the EGFR activation signals are hereby explicitly detailed, aims to translate the effective drug concentration into treatment efficacy. The PBPK models developed are independent from the disease model and have the objective of predicting the effective drug concentration evolution after its administration. The effective concentration corresponds to the unbound intracellular concentration (not linked with any protein other than EGFR) of the drug in tumor cells. It is presumed to be proportional to the unbound concentration of the drug in the organ in which the tumor is located. The two EGFR-TKI models built use the same model structure, whether it be for the EGFR inhibition or the pharmacokinetics (this structure could also be used to model EGFR-TKIs other than gefitinib or osimertinib). However, some elements still allow these two models to be distinguished: the inhibition constant for ATP binding in the tumor site (Table 2) and the PK data on which the models are calibrated (Section 2.4). For differentiating osimertinib and gefitinib, the mechanisms that will grant the tumor resistance to the treatment are more important.

### 2.3. Mechanisms of Resistance

In most cases, the development of a resistance mechanism is marked by a regrowth of the tumor after a phase of decline. Cancer drug resistance is viewed as based upon a plethora of distinct mechanisms. Drug resistance mutations can occur in the same protein or in different proteins, as well as in the same pathway or in parallel pathways, bypassing the mechanism of action of the drug. Neither all the genomic alterations nor the adaptations in the tumor microenvironment that generate resistance are known. Whether those mechanisms of resistance appear during the treatment or are already in place but only for a limited number of cells which are then selected by the treatment is uncertain and may depend on each patient and each mechanism of resistance [18,19]. In our case, for simplification purposes, we chose the second option and assumed that a limited number of cells bearing resistance mechanisms are already present in the tumor when starting the treatment. Gefitinib and osimertinib, even though they have a similar mechanism of action, have shared but also distinct mechanisms of resistance [20]. What also differentiates the two treatments is the frequency of apparition of the mechanisms of resistance, which can also be impacted by whether the treatment is given in the first or second line [21,22]. For each treatment, we have implemented the most prevalent associated mechanisms of resistance, namely, *EGFR* alteration (amplification, *T790M* mutation loss and *C797S* mutation), *MET* amplification, *KRAS* mutation or amplification and *PIK3CA* mutation or amplification for osimertinib [21] and *T790M* EGFR mutation and *MET* amplification for gefitinib (listed in decreasing order of frequency) [23]. Note that some mechanisms of resistance to osimertinib also provide resistance to gefitinib treatment, but they were not implemented in the model as their prevalence is negligible compared to the other resistance mechanisms to gefitinib. All this explains why our models do not integrate pathways or phenomena other than those representing the drug mechanisms of action and cancer growth shown in Figure 1.

In order to be able to represent both the tumor heterogeneity and the dynamics of resistance to a treatment, each tumor is divided into separate subclones that are distinct subpopulations of tumor cells, each carrying their own set of mutations. Among those subclones is a resistant subclone that can carry the resistance mechanisms described above. This subclone is initialized with a negligible size compared to other subclones; however, if it carries a mechanism of resistance, it will grow to the point where it will predominate over the other subclones, which, in contrast, will decline in response to the treatment. To reproduce the variability of response to EGFR-TKIs, more than 70 patient descriptors were introduced, with the complete list of sources available in the Appendix A (spreadsheet). As an example, regarding the resistance to treatment, for each virtual patient, the presence or absence of each resistance mechanism in the resistant subclone is drawn from a Bernoulli distribution whose parameter has been informed by the literature [21,23,24,25]. It is important to note that the presence of a resistance mechanism in the primary tumor does not necessarily mean that all the metastases will carry this resistance. Therefore, we introduced a parameter that either ensures the same mechanism of resistance in all the tumors and metastases (with a 70% probability) or allows for independent mechanisms of resistance between the primary tumor and the metastases (with a 30% probability). The assumption that, in 70% of *EGFR*-mutated LUAD cases, the mechanisms of resistance are the same in the tumor and metastases has been made based on discussions with clinicians.

### 2.4. Data for Model Calibration

Table 3 describes the datasets used to inform the model of the drug pharmacokinetics and mechanism of action.

As the available data were heterogeneous (from the Ki presented in Table 2 to the evolution of tumor size in xenograft mice), we applied a calibration strategy detailed by Palgen et al. [35]. This step-by-step approach minimizes an objective function to calibrate complex mechanistic models focusing on adjusting parameters that are challenging to derive from the literature. This method applies computational constraints derived from various sources such as textual descriptions, numerical values or graphical data, here listed in Table 3, combined with a CMAES algorithm, to ensure accurate replication of the desired pathophysiology within the specified context of use. Finally, to successfully calibrate the efficacy in humans using FLAURA and NEJ002 data (aggregated TTP in human population), we inferred distributions of parameters characterizing the virtual population, as realized by L’Hostis et al. [8]. In order to assess the model’s capacity to successfully reproduce EGFR-TKIs’ PK, or EGFR-TKIs’ efficacy in mice, we compared the data used for calibration to the model outputs.

### 2.5. Comparison of Model Prediction with a Retrospective Study

The retrospective study presented by Li et al. [36] compares the efficacy of osimertinib and gefitinib and was not used for calibration purposes. It was therefore possible to use this study to compare the model predictions with real-life data. This study included 102 patients with stage III-B or IV NSCLC, of whom 49 were treated with osimertinib and 53 with gefitinib. The PFS was measured and serves as a comparative endpoint for our model. Table 4 summarizes the characteristics of the patients that participated in this study and that were used to create the associated virtual population.

As we were comparing two distinct endpoints (PFS for retrospective study vs. simulated TTP from the ISELA-V2 model), we did not use traditional statistical comparison methods. Our objective was to ascertain the consistency of the outcomes, ensuring that the model’s predictions aligned with the real-world data observed in the study.

### 2.6. Effect Model

Lastly, we performed an in silico clinical trial to compare the efficacy of gefitinib and osimertinib in the same virtual population. By doing so, we could assess for each patient his time to progression, whether he is treated with osimertinib or with gefitinib, and plot the distribution of absolute benefit, i.e., the difference between the two scenarios. Table 5 contains the distribution for each of the patient’s descriptors used to generate a virtual population of *EGFR*-mutated advanced LUAD patients. The number of virtual patients has been defined to be 10 times the number of patients in the retrospective study in order to have enough variability in the virtual population as well as enough statistical units for bootstrapped approaches. This method has been described by Boissel et al. as a way to give insights on treatment efficacy and best responders characteristics [37].

## 3. Results

### 3.1. Reproducing Pharmacokinetic Data of Gefitinib and Osimertinib

After calibration, the PBPK models of gefitinib and osimertinib were able to reproduce the plasmatic concentrations of the drugs both in mice and in humans (Figure 2). All the simulated data points were within the range defined by the standard deviation.

### 3.2. Reproducing Epidermal Growth Factor Receptor Tyrosine Kinase Inhibitor Efficacy

After calibration, the model was also able to reproduce the tumor volume evolution after gefitinib or osimertinib treatment in mice transplanted with xenografts bearing an *EGFR* exon 19 deletion (Figure 3) and also with xenografts bearing both an *EGFR* exon 19 deletion and a *PIK3CA* mutation (Palgen et al. [35]). The model also reproduced the dynamics of the tumor without treatment (Palgen et al. [35]), ensuring that the treatment was indeed responsible for the shrinkage of the tumor when administered. Table 6 summarizes the calibration results in mice.

We used target population characteristics including age, gender, initial cancer stage and type of *EGFR* mutation to generate the virtual populations for FLAURA, NEJ002 and AURA3 clinical trials. Trial simulations on these virtual populations successfully reproduced the probability of tumor progression as measured in the real trials. In order to achieve these results, several parameters of the model that could not be informed by in vitro experiments or via animal testing were calibrated: the initial size of the micrometastases, the initial size of the resistant subclone, the reduced impact of the immune system in the metastases compared to the primary tumor [38] and the scaling factor regulating the distribution of the treatments in the tumors. To compare the goodness of fit of the simulated Kaplan–Meier curve with the one from the associated clinical trial, we used the MaxCombo method on R software, v.4. This approach is based on the use of a combination of weighted log-rank tests. The latter was integrated into a bootstrapped approach where 5000 randomly drawn samples, each containing a tenth of the global virtual population, were compared to the real-life observations. This statistical methodology has been described by Jacob et al. [34]. As illustrated in Figure 4, the threshold of the ratio of non-significant tests required to consider the two curves as similar was set to 80% and was largely exceeded for the two simulations (99.86% for the population treated with osimertinib and 98.42% for the population treated with gefitinib). The minor discrepancies can be attributed to the random sampling process; nevertheless, the obtained results show that, globally, the model successfully reproduces real-life results at a population level. More precise results could have been obtained if individual data were available.

### 3.3. Reproducing the Results from a Retrospective Study

We simulated a clinical trial comparing the efficacy of gefitinib and osimertinib in treating EGFR-mutated LUAD with virtual patients representing the participants of the retrospective study by Li et al. [36], as illustrated in Figure 5. In contrast to the previous simulation, no parameter was changed to fit the outcome of the retrospective study; only the patients’ characteristics were updated to match the ones of the real population described in the Li et al., study.

As presented in Table 7, the results of the in silico clinical trial show that osimertinib is associated with a longer TTP than gefitinib for the designed population with a median TTP of 20 months (95%CI: 15–24) for osimertinib and 11 months (95%CI: 7.5–12) for gefitinib. The retrospective study reported a median PFS of 18.1 months (95%CI: 15.4–20.7) for osimertinib and 10.7 months (95%CI: 9.9–11.4) for gefitinib, consistent with the TTP results of our in silico clinical trial, although the endpoints are not identical as the model predicts the TTP while the retrospective study reports the PFS. As, theoretically, the TTP cannot be lower than the PFS because the event of progression is part of the PFS, it is consistent that the simulated TTP is greater than the observed PFS. This result highlights the robustness of the model in predicting the efficacy of osimertinib and gefitinib in *EGFR*-mutated LUAD patients.

### 3.4. Effect Model

We applied an in silico effect model to the virtual population mentioned in the Materials and Methods section, shown in Table 4. While this method is exploratory due to the fact that the model is limited to population-level predictions, the insights one can deduce from such additional explorations underscore mechanistic models’ potential.

By simulating the time to progression for each patient of the virtual population, whether they are treated with osimertinib or gefitinib, we can directly compare the efficacy of the two treatments by plotting an effect model (Figure 6). This effect model suggests that treatment with osimertinib is linked to a longer TTP for the majority of patients. This result is even more greatly accentuated for patients carrying an *EGFR* exon 20 mutation, which should, however, be interpreted with caution as the data that were used to calibrate the model did not contain any patients carrying *EGFR* exon 20 mutations. Only patients who progressed when treated with both gefitinib and osimertinib were included in the effect model, but it is interesting to note that, out of the 1020 virtual patients, 329 did not progress during the 24 months of simulation when treated with osimertinib, while there were only 119 for gefitinib. This is additional information supporting the superior efficacy of osimertinib. Furthermore, it shows that one can make individual predictions and use the model to figure out factors of response. This exploration serves to broaden possibilities, and future calibration with individual patient data would enhance the precision of our findings and their applicability to personalized treatment strategies.

## 4. Discussion and Conclusions

The success of the calibration in reproducing the pharmacokinetics and drug effect of gefitinib and osimertinib in mice and humans provides evidence of the robustness, and, by extension, the credibility, of the model [39]. The model credibility is further enhanced by the reproduction of a retrospective study that contains patient data not used during the calibration process using only the patient characteristics at the population level. This provides evidence that knowledge-based mechanistic models can comprehend the complexity of biological phenomena to predict the result of experiments or clinical trials by integrating already gathered knowledge and data. Such models could play a major role in the development of new therapies [40]. With this credible model, we showed that one can predict which of two treatments is more efficacious at a population level. Its extension could be used to identify patients who would respond the best to a given treatment. Another attractive aspect of such models is their ability to generate control arms for real-life clinical trials. Indeed, our model successfully predicted responses to osimertinib and gefitinib, two standards of care, in an EGFR-mutated NSCLC population. In order to achieve this impression of a real trial, we generated virtual patients representative of the target trial population. Following a similar process, we could create a virtual control arm with simulated patients receiving the standard of care. This virtual control arm could then be compared to any arm of real patients receiving an investigational treatment. This comparison allows for a more accurate assessment of the investigational treatment’s efficacy because the generated virtual population enrolled in the control arm has the same measurable characteristics as the group from the investigational arm. In addition, the construct of the virtual arm can account for the unmeasurable patient characteristics that might alter the difference in investigational and control drug efficacies. This asset can only be found with numerical methods and is particularly useful in the case of rare diseases where the recruitment of patients can be difficult. While potentially yielding amazing perspectives, such approaches should be considered as explorations because, as of today, due to the domain of application of the model, they are limited to population-level predictions [39]. While the model highlights promising applications, it also opens avenues for further enhancements. Currently, ISELA-V2’s scope does not extend to modeling the drug’s adverse effects or the patient’s health status, key factors in predicting mortality. Integrating specific adverse effects into the model is a potential relevant improvement, although a comprehensive mechanistic model of patient death may still be beyond reach. Here, statistical models could complement our approach, offering a viable strategy for predicting death. Furthermore, this limitation becomes easier to manage when the safety profile of a treatment is well established, potentially allowing our model to contribute valuable insights into treatment outcomes without directly modeling overall survival rates.

Moreover, the ambition to model the impact of patient-specific conditions on drug pharmacokinetics such as renal impairment highlights a relevant potential future direction [41]. Despite the challenge posed by the current availability of data, which predominantly focus on population-level characteristics, advancing towards individualized predictions based on specific patient attributes remains a key goal.

Nevertheless, the model presented in this article could become an asset in the search for new therapeutic strategies in *EGFR*-mutated lung cancer. It has even recently been utilized to successfully predict the results of Phase III clinical trials in the FLAURA2 and MARIPOSA trials (94.0 and 98.1% of non-significant, bootstrapped log-rank tests, respectively [42,43]), further enhancing its credibility. This could help guide the recruitment of patients for clinical trials, recruitment whose quality dictates the outcome of the trial and that can be challenging depending on the rarity of the disease [44].

## Figures and Tables

**Figure 1 biomedicines-12-00704-f001:**
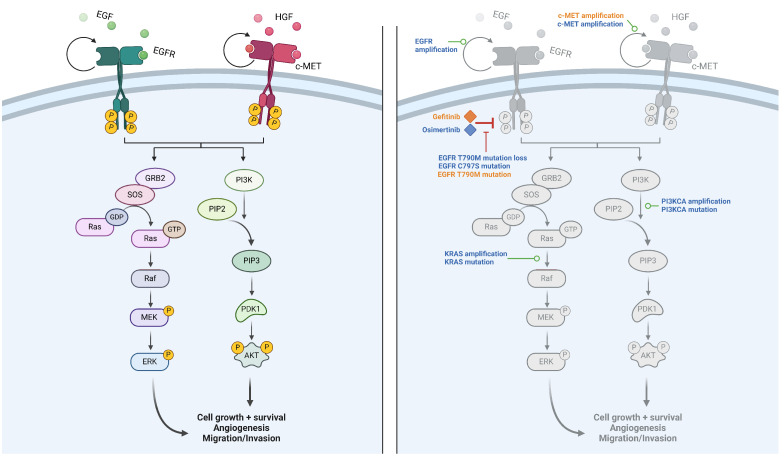
Downstream signaling pathways and resistance mechanisms. Left: downstream pathways following EGFR or MET activation, either after activation by EGF or HGF or via constitutive activation caused by a mutation. VEGFR is not shown in the figure but also activates the same pathways. Right: resistance mechanisms to osimertinib (blue) and gefitinib (orange). Created with Biorender.com, 2024.

**Figure 2 biomedicines-12-00704-f002:**
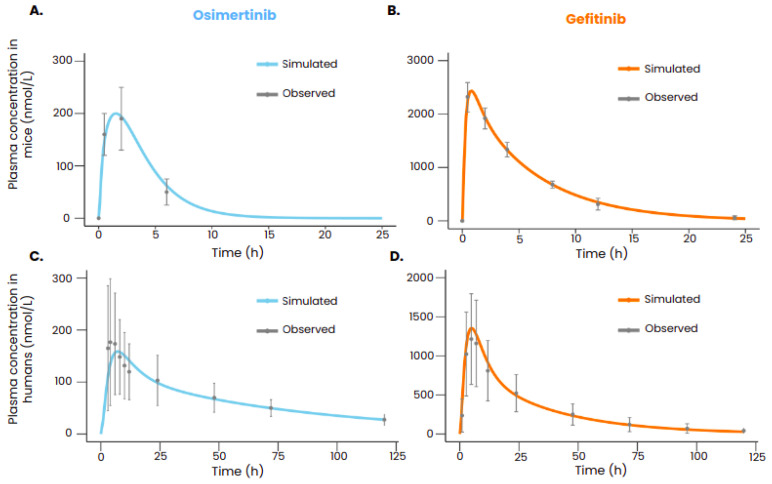
TKI plasma concentration in mice after an oral administration of 5 mg/kg of osimertinib (**A**) or 50 mg/kg of gefitinib (**B**). TKI plasma concentration in humans after an oral administration of 80 mg of osimertinib (**C**) or 250 mg of gefitinib (**D**). The observed values with the standard deviation are represented in gray ((**A**): Yates et al. [28], (**B**): Wang et al. [27], (**C**): Zhao et al. [30], (**D**): Bergman et al. [29]), and the simulated plasmatic concentration of the drug with the PBPK models is represented in color. The administration was given at t = 0.

**Figure 3 biomedicines-12-00704-f003:**
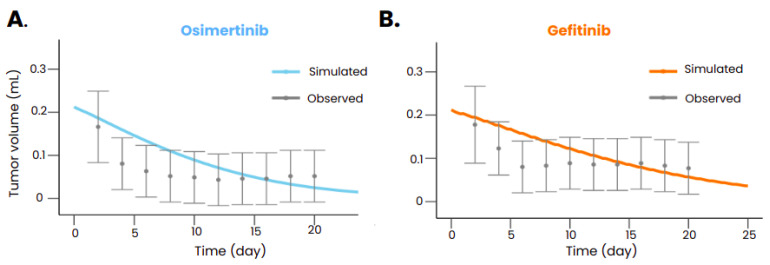
Tumor volume evolution in mice implanted with a tumor carrying an exon 19 deletion mutation and treated with osimertinib (**A**) and gefitinib (**B**). The observed data (Kang et al. [26]) with the standard deviation are represented in gray, and the tumor volume predicted with the models is represented in color. Drugs were administered orally starting at t = 0 with a posology of 25 mg/kg daily for gefitinib and 6.25 mg/kg daily for osimertinib.

**Figure 4 biomedicines-12-00704-f004:**
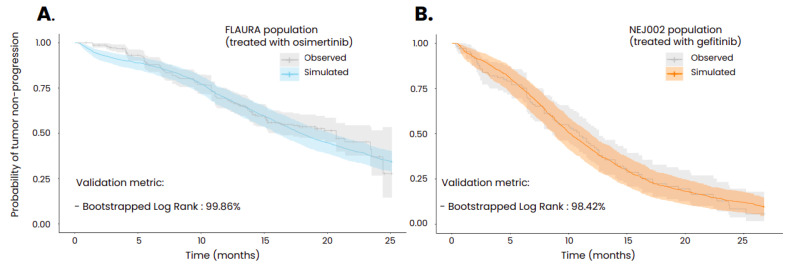
Kaplan–Meier curves of time to progression obtained in the in silico clinical trials performed with the ISELA-V2 model and compared to real clinical trial data ((**A**) FLAURA clinical trial; (**B**) NEJ002 clinical trial). The uncertainty interval of the simulated curve corresponds to the variability obtained from bootstrapping the virtual population, while the 1 of the observed curves stands for the 95% confidence interval. In the FLAURA study, patients were treated with 80 mg of osimertinib administered orally daily, and, in the NEJ002 study, patients were treated with 250 mg of gefitinib orally daily. Both treatments started at t = 0.

**Figure 5 biomedicines-12-00704-f005:**
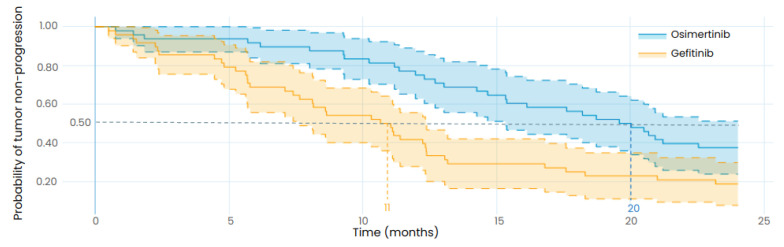
Exploratory in silico clinical trial: Comparison of gefitinib and osimertinib performed with the LUAD model and with the same patient baseline characteristics as the retrospective study from Li et al. The uncertainty intervals stand for the 95% confidence intervals of the progression curves. The dotted lines represent the median TTP. Patients were orally treated with 250 mg of gefitinib daily in the gefitinib arm (in orange) and with 80 mg of osimertinib daily in the osimertinib arm (in blue). Both treatments were started at t = 0.

**Figure 6 biomedicines-12-00704-f006:**
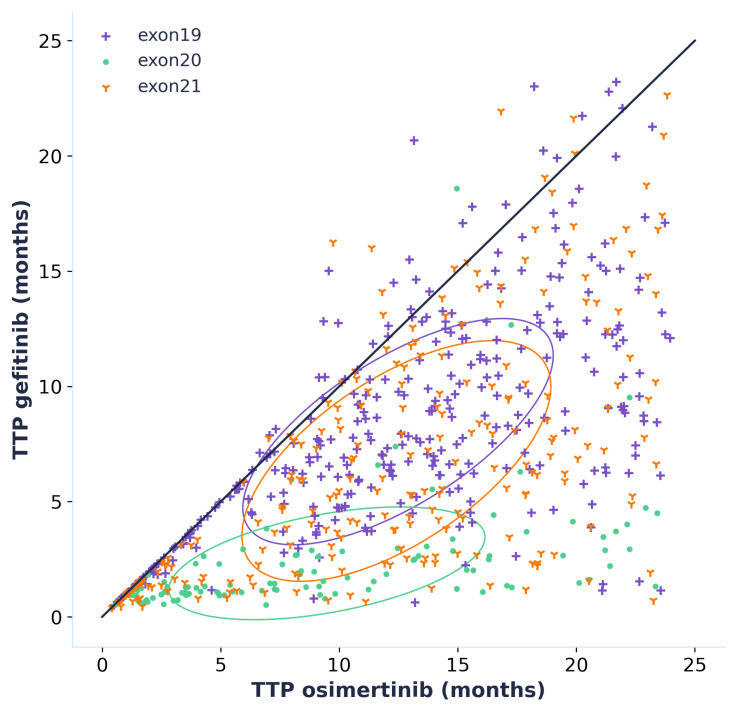
Effect model comparing the effect of osimertinib and gefitinib in terms of time to progression (TTP) in a virtual population. Only virtual patients that progressed within the 24 months of the in silico trial with both osimertinib and gefitinib are plotted (691 out of the 1020 that were simulated). Each point corresponds to one virtual patient defined by time to progression when treated with osimertinib and when treated with gefitinib. The circled areas correspond to confidence ellipses computed using the Pearson coefficient. (the number of standard deviations was set to 1).

**Table 1 biomedicines-12-00704-t001:** Parameters used to model the activation of RAS and PI3K.

Parameter Name	Description	Value
kaPi3kByMet	Forward speed of PI3K activation by MET	$4.69\times 10^{6}$ (L/mole/s)
kaPi3kByVegfr	Forward speed of PI3K activation by VEGFR	$9.99\times 10^{7}$ (L/mole/s)
kaPi3kByEGFR	Forward speed of PI3K activation by EGFR	$6.04\times 10^{5}$ (L/mole/s)
kaPi3kByras	Forward speed of PI3K activation by RAS	$3.94\times 10^{4}$ (L/mole/s)
kdaPi3k	Reverse speed of PI3K activation	1 (1/s)
kaRasByMet	Forward speed of RAS activation by MET	$1.89\times 10^{6}$ (L/mole/s)
kaRasByVegfr	Forward speed of RAS activation by VEGFR	$1.34\times 10^{3}$ (L/mole/s)
kaRasByEGFR	Forward speed of RAS activation by EGFR	$1.89\times 10^{6}$ (L/mole/s)
kdaRas	Reverse speed of RAS activation	1 (1/s)

**Table 2 biomedicines-12-00704-t002:** Values of the inhibition constant of gefitinib and osimertinib for the binding of ATP to the intracellular domain of EGFR depending on the *EGFR* mutation. Values have been taken from or computed from Eck et al. [15] (gefitinib) and Masuzawa et al. [16] and the EMA report [17] (osimertinib). Note: AZ5104, the most active metabolite of osimertinib that was included in the model, has been assumed to have the same Ki values as osimertinib.

*EGFR* Mutation	Tyrosine Kinase Inhibitor	${\mathrm{Ki}}_{\mathrm{TKI}}$ (nmol/L)	$\frac{{\mathrm{KI}}_{\mathrm{TKI}}}{\mathrm{atpAffinity}}\times 10^{-3}$
Exon 19 deletion + *T790M* resistance mutation	Gefitnib	4.3×10	5.1
	Osimertinib	$3.2\times 10^{-2}$	$3.8\times 10^{-3}$
Exon 21 insertion + *T790M* resistance mutation	Gefitinib	$2.9\times 10^{1}$	3.4
	Osimertinib	$9.4\times 10^{-3}$	$1.1\times 10^{-3}$
Exon 19 deletion mutation	Gefitnib	$8.3\times 10^{1}$	$6.5\times 10^{-3}$
	Osimertinib	1.1	$8.5\times 10^{-3}$
Exon 20 sensitive mutation	Gefitinib	2.5	$1.3\times 10^{-1}$
	Osimertinib	$5.9\times 10^{-1}$	$3.1\times 10^{-2}$
Exon 20 resistant mutation	Gefitnib	2.6×10	$7.0\times 10^{-1}$
	Osimertinib	6.2	$1.7\times 10^{-1}$
Exon 21 insertion mutation	Gefitinib	6.4	$4.3\times 10^{-2}$
	Osimertinib	$9.7\times 10^{-1}$	$6.5\times 10^{-3}$
Wild type	Gefitnib	1.6×10	3.2
	Osimertinib	3.3	$6.4\times 10^{-1}$

**Table 3 biomedicines-12-00704-t003:** Data used during the calibration of the model for the pharmacokinetics of the drugs and their mechanism of action.

Study	Biological Process to Reproduce	Experimental Conditions	Treatment	Type of Study
Kang et al. [26]	Tumor volume evolution after treatment administration in mice	Mouse PDX model YHIM-1003 harbors *EGFR* exon 19 deletion. Mouse PDX model YHIM-1009 harbors *EGFR* exon 19 deletion and *PIK3CA E542K* mutation. Treated with gefitinib or osimertinib	Gefitinib, osimertinib	Pre-clinical
Wang et al. [27]	Gefitinib plasmatic profile in mice	Tumor bearing mice that have been administered 50 mg/kg of gefitinib orally	Gefitinib	Pre-clinical
Yates et al. [28]	Osimertinib and AZ5104 plasmatic profile in mice	Tumor-bearing mice that have been administered 5 mg/kg of osimertinib orally	Osimertinib	Pre-clinical
Bergman et al. [29]	Gefitinib plasmatic profile in humans	Healthy volunteers who have been administered 250 mg of gefitinib orally	Gefitinib	Clinical
Zhao et al. [30]	Osimertinib and AZ5104 plasmatic profile in humans	*EGFR*-mutated NSCLC patients who have been administered 40 mg or 80 mg of osimertinib orally	Osimertinib	Clinical
FLAURA [31] ^2^	Distribution of time to progression (computed from OS and PFS curves ^1^) in target population + distribution of progression cause and site of new lesions	Patients with an advanced stage of NSCLC harboring an *EGFR* mutation, treated as first line with 80 mg/day of osimertinib	Osimertinib	Clinical
NEJ002 [32] ^2^	Distribution of time to progression (computed from OS and PFS curves ^1^) in target population	Patients with an advanced stage of NSCLC harboring an *EGFR* mutation, treated as first line with 250 mg/day of gefitinib	Gefitnib	Clinical
AURA3 [33] ^2^	Distribution of time to progression (computed from OS and PFS curves ^1^) in target population + distribution of progression cause and site of new lesions	Patients with *EGFR T790M*-positive advanced-stage NSCLC who previously had disease progression during first-line EGFR-TKI therapy, treated with 80 mg/day of osimertinib	Osimertinib	Clinical

^1^ The methodology used to determine the time to progression is explained by Jacob et al. [34]. ^2^ Population-level data were used as we did not have access to individual patient data.

**Table 4 biomedicines-12-00704-t004:** Characteristics of the patients from the retrospective study that were used to create the related virtual population. Values taken from Li et al. [36].

	Treated with Osimertinib (n = 49)	Treated with Gefitinib (n = 53)
Sex (M/F)	24/25	26/27
Age (<65/>65)	27/22	25/28
Smoking status (Y/N)	38/11	35/18
Cancer stage (IIIb/IV)	30/19	35/18

**Table 5 biomedicines-12-00704-t005:** Characteristics of the virtual population used to make the effect model.

	Virtual Population
Sex (M:F ratio)	1:2
Age (mean, sd)	(67, 11)
Smoking status (Never, Former, Current)	(28%, 34%, 38%)
Ethnicity (Asian, Other)	(55%, 45%)
*EGFR* mutation (19, 20, 21)	(51.6%, 13.2%, 35.2%)

**Table 6 biomedicines-12-00704-t006:** Calibration results for the efficacy of osimertinib and gefitinib based on mice experiments (Kang et al. [26]).

*EGFR* Mutation	Ratio of Points within the Standard Deviation	Gefitinib	Osimertinib
Exon 19 deletion mutation	100%	90%	80%
Exon 19 deletion mutation + *PI3KCA* mutation	80%	100%	100%

**Table 7 biomedicines-12-00704-t007:** Results from the retrospective study by Li et al. [36] and from the in silico clinical trial performed with the ISELA-V2 with the same patient characteristics as the retrospective study.

	Retrospective Study	In Silico Clinical Trial with the Same Patient Characteristics
Osimertinib	PFS: 18.1 months (95%CI: 15.4–20.7)	TTP: 20 months (95%CI: 15–24)
Gefitnib	PFS: 10.7 months (95%CI: 9.9–11.4)	TTP: 11 months (95%CI: 7.5–12)

## Data Availability

The datasets generated and/or analyzed during the current study are available from the corresponding author on reasonable request. The code utilized during the current study can be made available on the jinkō.ai platform by the corresponding author on reasonable request.

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
