# Peer review of "Comparing the Efficacy of Two Generations of EGFR-TKIs: An Integrated Drug–Disease Mechanistic Model Approach in EGFR-Mutated Lung Adenocarcinoma"

_biomedicines, 2024, doi:10.3390/biomedicines12030704_

Round 1

Reviewer 1 Report

Comments and Suggestions for Authors

Overall this is an interesting paper.

However there is a major biological consideration and are several suggestions for improvement:

From a biological point of view it is wrong to state that Gefitinib and Osimertinib have an identical mechanism of action. They both inhibit EGFR but there are several differences. This cannot be dysregarded, so authors are encouraged to carefully check on that.

Besides this major biological issue, which is super important, these are several suggestions for improvement. These are not corrections but really suggestions to improve and make the paper easier to comprehend.

-Acknowledge the significance of patient heterogeneity at the population level and its impact on treatment responses in NSCLC. Highlight the role of tumor heterogeneity in predicting treatment outcomes and guiding personalized treatment strategies. Discuss what models can provide in this regard. Emphasize the potential of mechanistic models in personalized medicine, patient selection for clinical trials, and optimizing therapeutic interventions based on patient characteristics. Clearly state how the models address existing gaps in knowledge or practice.

-Discuss the importance of generating virtual control arms for clinical trials using mechanistic models to assess investigational treatment efficacy accurately, particularly in rare diseases (recurrence may be eventually considered a rare disease as tumors relapse differently)

-Address limitations of the model, such as the inability to predict patient death and the impact of cancer-related symptoms on drug pharmacokinetics, and suggest further development to enhance predictive capabilities.

-The abstract is quite lengthy and detailed, which may make it challenging for readers to grasp the key points quickly. Consider condensing the language to ensure clarity and conciseness, focusing on the most critical aspects of the study.

-The abstract mentions that the model replicated the time to progression observed in clinical trials and was able to reproduce results from a retrospective study. It would be beneficial to provide more specific quantitative results or metrics to demonstrate the model's accuracy and performance described in visuals.

-Provide more details on how the KBM and pharmacokinetic models are integrated to simulate patient responses to different EGFR TKIs. Explain the rationale behind combining these models and how they complement each other in analyzing treatment outcomes and efficacy.

-Clearly state the objectives of the study in relation to the development and application of the ISELA-V2 model. Explain how the study aims to use this model to compare treatment outcomes and guide more effective clinical decision-making for LUAD patients.

-The description of the construction of the ISELA-V2 model could be more structured and explicit. Provide a clear outline of the steps involved in improving the disease model and integrating it with the PBPK treatment models for gefitinib and osimertinib.

-While the development of PBPK models for both gefitinib and osimertinib is discussed, consider providing more detailed explanations of the underlying assumptions, equations, and physiological parameters used in the models. Additionally, clarify how the PBPK models were validated and optimized to ensure their accuracy in predicting drug concentrations.

-The section on modeling TKI mechanism of action and mechanisms of resistance is thorough, but complex. Consider simplifying the explanation by breaking down the equations and processes into more digestible components. Additionally, provide visual aids or diagrams to help illustrate the signaling pathways and interactions between different components.

-The description of the data used for model calibration is brief. Provide more information on the sources of the calibration data, the specific parameters being calibrated, and the methodology used for optimization. Transparency in model calibration is essential for ensuring the validity and reliability of the model predictions.

-Enhance the description of how the model predictions were compared with the retrospective study data. Provide details on the specific metrics or statistical methods used to assess the accuracy of the model predictions against the real-life data from the study. Explain any discrepancies between the model predictions and real-life data and discuss the implications of these findings for the efficacy of gefitinib and osimertinib in treating EGFR mutated LUAD.

-Elaborate on the use of bootstrapped approaches in analyzing the in silico clinical trial results. Explain how bootstrapping was implemented, its advantages in providing insights on treatment efficacy and identifying best responders, and any potential limitations or considerations in the approach.

Comments on the Quality of English Language

good english

however make it more concise

Reviewer 2 Report

Comments and Suggestions for Authors

This paper is about modelling  the pharmacokinetics and drug effect of gefitinib and osimertinib in mice and humans Using data from clinical trials of gefitinib and osimertinib a virtual/synthetic cohort is generated. The modelling work is well described but what i miss is the clinical implications of the model. I think also the source of data is difficult to understand and could be more clearly described. An obvious weakness is the complexity of resistance, down stream signalling and crosstalk. Which is only partly addressed in the models. Even though the authors claim that the models can could be used to identify patients that would respond it is not clear in what way (which factors).  All in all the study is more of an exersise and clinical impact is probably limited.

Round 2

Reviewer 1 Report

Comments and Suggestions for Authors

The manuscript has improved significantly and I commend the authors for this.

However, there are some aspects that could be improved, such as: the phrasing is too long and dense in some parts, and could be shortened and simplified to make it more readable and accessible. Some sentences could be rephrased or merged to avoid repetition and redundancy.

The manuscript is still not perfectly written and edited, and contains some errors and inconsistencies in grammar, spelling, punctuation, and formatting, despite several xx that have not been properly filled in (also in the letter).

These inconsistencies could affect the clarity of the manuscript, which is otherwise interesting. Therefore, it should be corrected and revised.

Comments on the Quality of English Language

Please see above.

Reviewer 2 Report

Comments and Suggestions for Authors

The manuscript have improved and the comments by reviewers have been adequately addressed.

Author Response

Dear reviewer,

On behalf of all the co-authors, I would like to thank you for the reviews of this article as well as for all these constructive comments. We all believe that these comments helped us make the paper better and more understandable for all the journal's readers.

Best regards, Claudio Monteiro